# Microstructural Heterogeneity and Property Variations in Cast and Vacuum Hot-Pressed CoCrPtB Alloy

**DOI:** 10.3390/ma17030544

**Published:** 2024-01-23

**Authors:** Yan Li, Weiming Guan, Ming Wen, Junmei Guo, Ze Chen, Chuanjun Wang

**Affiliations:** 1Yunnan Precious Metals Laboratory, Sino-Platinum Metals Co., Ltd., Kunming 650106, China; lyan202311@126.com (Y.L.);; 2Kunming Metallurgy College, Kunming 650033, China; 3School of Mechanical and Aerospace Engineering, Nanyang Technological University, 50 Nanyang Avenue, Singapore 639798, Singapore

**Keywords:** CoCrPtB alloy, vacuum hot pressing, secondary dendrite arm spacing, magnetic materials, hysteresis loop

## Abstract

Limited research has been undertaken regarding the homogeneity of CoCrPtB alloy billets. A CoCrPtB alloy was processed through casting and vacuum hot pressing. This investigation delved into the interconnection between the secondary dendrite arm spacing (SDAS) in the as-hot-pressed samples and their corresponding attributes, specifically Vickers hardness and magnetic properties. Systematic sampling was conducted on the cross-sectional layer and longitudinal surface. Upon examination of the cross-sectional layer proximate to the uppermost region of the hot casting, a discernible parabolic trend was observed for the SDAS that exhibited a gradual increment from the peripheral regions toward the central area along the width. Simultaneously, the fraction of the dendrite phase displayed a consistent linear decline, attaining its peak value at the central portion of the billet. Conversely, on the longitudinal surface, SDAS and the fraction of the dendrite phase remained fairly uniform within the same column sampling regions. However, a notable divergence was identified in the central section, characterized by an augmented SDAS and diminished dendrite phase content. This inherent microstructural inhomogeneity within the CoCrPtB alloy engendered discernible disparities in material properties.

## 1. Introduction

The fundamental premise that underlies the utilization of materials hinges on the ability to manipulate their microstructural features and, subsequently, their ensuing properties. In the context of numerous alloy materials, the formation of dendritic structures is a common occurrence during the casting process. Regrettably, these structures often prove resilient even when subjected to thermal-mechanical treatments such as forging, rolling, and heat treatment. Consequently, a well-established body of research exists dedicated to exploring the relationship between dendritic structures and the properties of materials [1,2,3,4,5,6,7,8,9,10,11]. Secondary dendrite arm spacing (SDAS, $\lambda_{2}$) is commonly used to observe casting and the ensuing manufacturing processes, primarily due to its significant influence on the mechanical and magnetic characteristics of magnetic alloys [3]. For example, alloy or trace element adding can reduce the size of SDAS [1,2]; local solidification time is crucial for the cooling rates and affects the SDAS size and distribution of casting alloys [5]. Besides, many methods have been investigated to reasonably determine the SDAS and its relation to the properties of materials [8,10,11]. Concurrently, SDAS is intricately associated with the solidification mechanisms [3,4], growth patterns [7,9], and distribution of solute, inclusions, and interfacial phases within the material matrix. The exploration of the intricate interplay between SDAS and the conditions governing the solidification process offers a twofold advantage: it facilitates the prediction of the microstructure in the as-cast state and enhances the alloy’s performance through subsequent processing stages.

CoCrPtB alloy sputtering plays a pivotal role in producing low-anisotropy or capped layers [12] in perpendicular recording media, especially within the hard disk industry. Numerous scholarly works have shed light on the methodologies employed in the fabrication of this particular alloy [13,14,15]. Cobalt (Co) serves as the primary contributor to its ferromagnetic properties; a higher Co content results in an alloy with elevated saturation magnetization (M_s_). Platinum (Pt) is incorporated to enhance the magneto-crystalline anisotropy constant K_u_, consequently increasing the coercivity H_c_ of the cobalt-based alloy. Chromium (Cr) is introduced to improve corrosion resistance and tends to segregate at the boundaries of the Co alloy, reducing exchange coupling between grains and subsequently minimizing media noise. Furthermore, the addition of Boron (B) achieves grain isolation and reduces grain sizes [16]. Therefore, the uniformity of the CoCrPtB alloy is paramount for the consistent production of media films. Typically, this alloy is synthesized through casting, followed by processes such as hot isostatic pressing, hot pressing, rolling, and heat treatment. These steps are undertaken to rectify inherent as-cast defects such as shrinkage and porosity and to ensure the overall homogenization of the alloy. However, there is a notable scarcity of studies on the homogeneous conditions of the CoCrPtB alloy in its as-cast or as-pressed ingot state.

This investigation aims to explore the SDAS, Vickers hardness, and magnetic characteristics of a CoCrPtB alloy billet that has been meticulously synthesized through the casting and hot pressing processes. The primary objective is to elucidate the uniformity in microstructure and material properties and to establish a comprehensive understanding of their interdependence.

## 2. Materials and Methods

### 2.1. Materials and Experimental Parameters

The alloy composition in this study consists of CoCr_13_Pt_16_B_10_. To ensure the requisite level of alloy purity, raw materials comprising bulk Co (Jin Chuan Group. Beijing, China), Cr (Fdimaterials, Beijing,China), Pt (Sino-Platinum Metals Co., Ltd, Kunming, China), and B powders(ZNXC, Beijing, China), each with a purity exceeding 99.95%, were utilized. To mitigate the evaporation of B, a master alloy, CoB was meticulously prepared before initiating the alloy melt. The alloy ingot was fabricated through vacuum induction melting, employing a graphite mold.

During the melting and pouring procedures, a background vacuum level of 10 torr was maintained, with a maximum pour temperature of 1600 °C and a pour rate of 150 g/s. After pouring, the ingot was left in the chamber to cool for 1 h. This process yielded an ingot with a height of 120 mm, width of 100 mm, and thickness of 22.5 mm. To ameliorate the internal porosity inherent to casting, vacuum hot pressing was subsequently employed, specifically on the surface spanning 120 mm by 100 mm. Hot pressing was conducted at a sustained pressure of 40 MPa and a temperature of 1000 °C for 2 h (Vacuum Technology Co., Ltd., Shengyang, China).

### 2.2. Testing Method

The illustration depicting the sampling methodology is provided in Figure 1.
Metallography and hardness testing samples were taken on the margin ("margin" in Figure 1, row/column 1) and in the middle (“mid.” in Figure 1, row/column 2) on the cross-section (x-y quadrant) and longitudinal (i-j quadrant) surfaces.Magnetic performance testing samples were taken from the longitudinal surface with a thickness of 11 mm and cut in half from the center; exterior and interior regions were tested.Sample signs for metallography and hardness testing: S represents sample; superscript xy or ij denotes the sampling quadrant, the first subscript number represents the row or column, the second represents the sequence number, for example, $S_{2,4}^{ij}$ stands for the fourth sample in the middle column on the longitudinal surface.Sample signs for magnetic testing: the first subscript letter denotes the sampling position, and the second distinguishes the interior or exterior region; for example, $S_{D,E}^{ij}$ denotes the exterior region of sampling position Dd.

The major elements of CoCrPtB alloy were tested by inductively coupled plasma-optical emission spectrometer (ICP-OES, PE Optima 8300, USA, sourcing from Sino-Platinum Metals Co., Ltd.). The alloy was characterized through X-ray diffraction (XRD) employing an X-ray diffractometer (Rigaku, Japan, sourcing from Sino-Platinum Metals Co., Ltd.) with Cu radiation (l = 0.154056 nm) at a voltage of 45 kV and current of 40 mA. The XRD data was acquired at a scanning rate of 5°/min, with a step size of 0.02°, encompassing a 2θ range spanning from 20° to 100°. The software MDI Jade 6.0 was employed for data fitting and result computation. Scanning electron microscopy (SEM) equipped with an energy-dispersive X-ray spectroscopy (EDS) was conducted using Hitachi S-3400N SEM (Japan, sourcing from Sino-Platinum Metals Co., Ltd). Optical imagery was observed with a Leica DM4000 optical microscope (USA, sourcing from Sino-Platinum Metals Co., Ltd). SDAS and the dendrite arm area fraction were measured using professional image analysis software [17] (Image Pro Plus 6.0). To minimize the influence of possible measurement errors on the final result, an average value of at least 100 measures [1] was applied for the SDAS measurement per method D as Eli Vandersluis et al. proposed [17]. The fraction of the dendritic arm area was calculated for the primary and secondary dendrite arms. Vickers hardness was conducted on a Shimadzu HMV-FA2 Hardness tester for more than 3 spots. Magnetic measurements were performed using a NIM-2000H magnet property measurement system (National Institute of Metrology, Beijing, China). All micrographical and hardness testing specimens were prepared by manual grinding with Al_2_O_3_ paper and polished with different sizes of diamond abrasives (15 μm, 9 μm, 6 μm) successively. Finally, the excess abrasive particles were removed from the polished surfaces using an ultrasonic cleaner with acetone and water.

## 3. Results and Discussion

### 3.1. Composition and Phase Component

The chemical composition of CoCr_13_Pt_16_B_10_ alloy is listed in Table 1. The analyzed results accord well with the nominal composition.

Figure 2 shows the XRD patterns of CoCr_13_Pt_16_B_10_ alloys for the as-cast and as-hot-pressed samples.

It can be seen that both as-cast and as-hot pressed alloys have a face-centered cubic (fcc) structure (α-Co solution phase) and hexagonal close-packed (hcp) structure (ε-Co solution phase); no inter-metallic compound or single elemental phase was detected; Cr, Pt, and B are dissolved in Co solution [18]. The I_(002)_ peak of the ε-Co phase overlaps the I(111) peak of the α-Co phase. I_(200)_ and I_(101)_ are characteristic peaks that can be used to distinguish different Co phases [19]. Take I_(200)_ for α-Co, and I_(101)_ for ε-Co, the intensity ratio of I_fcc(200)_/I_hcp(101)_ = 1.88 for the as-cast ingot and 2.69 for the as-hot-pressed billet; in turn, the calculated percentage of the ε-Co phase from XRD patterns is 34.72% in the as-cast ingot and 27.10% after hot pressing. The hot pressing process decreases the fraction of ε-Co phase in the alloy.

Figure 3 shows the OM and SEM micrographs of as-cast and as-hot-pressed CoCr_13_Pt_16_B_10_ samples.

Figure 3 shows the OM and SEM micrographs of the as-cast and as-hot-pressed CoCr_13_Pt_16_B_10_ samples. As depicted in Figure 3a,b, the as-cast specimen exhibits a dendritic morphology characterized by the white–gray regions corresponding to dendrite arms (primary and secondary dendrites). The darker gray areas and certain adjacent small white–gray regions represent the inter-dendrite phase. Notably, the dendrite arm structure constitutes a substantial proportion of the microstructure. 

A quantitative assessment was carried out, involving the determination of the proportion of area occupied by these distinct regions within three separate fields of view in the same sample. The statistical analysis revealed an average proportion of 72.19% for the dendrite arm area. By comparing the XRD result and the statistical fraction of the dendrite arm area, the latter is close to the fraction calculated with I_fcc(200)_/I_hcp(101)_ in Figure 2. Additionally, the average SDAS value for the as-cast sample (sampling near the hot top area) was 12.05 μm. Conversely, the as-hot-pressed sample, depicted in Figure 3c,d, maintains its dendrite structural morphology. Notably, the proportion of dendrite arm structures further increases, averaging 80.01%, while the SDAS decreases to 8.51 μm.

Table 2 shows the EDS result regarding the distribution proportions of the major elements within the dendrite core (point 1, as indicated in Figure 3a,c) and the inter-dendrite area (point 2, as delineated in Figure 3b,d) following the casting and hot pressing processes.

The degree of segregation can be estimated using the subsequent equation [20]:(1)$$\kappa =C_{DC}/C_{ID}$$
where *C_DC_* and *C_ID_* are element content of the dendrite core and inter-dendrite region. When *κ* < 1, the element exhibits positive segregation between dendrites. The smaller the *κ*, the more severe the segregation. When *κ* > 1, the element exhibits negative segregation in the dendrite core; the larger the *κ*, the more severe the segregation. When *κ* = 1, there is no segregation of the element. From Table 2, it can be seen that the *κ* value of element Pt in the as-cast sample is 3.21, which is larger than 1, indicating negative segregation of the element in the dendrite core. The value of 0.99 for Cr is ~1, indicating relatively no Cr segregation. Compared with the as-cast results, the condition for the as-hot-pressed dendrite segregation is significantly improved, indicating that the hot-pressing parameters are insufficient to promote adequate diffusion of solute atoms, however, the average SDAS decreased. The measurement of SDAS serves as an indicator of the level of refinement within the dendrite structure. A reduced inter-dendrite spacing corresponds to a finer and more densely packed dendrite arrangement. Moreover, this reduction results in a narrower local distribution range of segregated elements. Consequently, following the hot pressing process, the extent of dendrite segregation within the CoCr_13_Pt_16_B_10_ alloy was ameliorated, positively influencing the material properties of the alloy ingot. Numerous research studies have substantiated the significant influence of SDAS on the mechanical characteristics of alloys [21]. It has been consistently observed that a finer SDAS correlates with enhanced processing performance in alloy materials.

Furthermore, the findings derived from XRD, OM, and SEM collectively reveal a noticeable shift in the microstructural composition post-hot-pressing. Specifically, there is an increase in the α-Co phase content while the ε-Co phase proportion diminishes. Broadly speaking, metals or alloys characterized by a face-centered cubic (fcc) crystalline structure, which boasts a total of 12 primary slip systems ({111}<110>), tend to exhibit superior processing characteristics compared to those with a hexagonal close-packed (hcp) structure, which is characterized by only three main slip systems ({0001}<11
$\bar{2}$
0>). A greater number of slip systems directly corresponds to an enhanced deformation coordination within the metal. This, in turn, results in reduced resistance to plastic deformation, rendering the material more amenable to processing. Consequently, the observed increase in the α-Co phase content after hot-pressing is conducive to the subsequent hot-rolling process.

In conjunction with the beneficial effects of adding Pt, Cr, and B to the CoCr_13_Pt_16_B_10_ alloy from an application perspective, this alloy modification elicits an augmentation in material strength and hardness, concomitant with reduced plasticity, ultimately leading to a detriment in its processing performance. The investigation revealed a proclivity for crack formation during and after the hot rolling process. Consequently, in order to enhance the effectiveness of the standard subsequent hot-rolling experiments, a more rigorous and systematic approach was employed. This involved extensive sampling and comprehensive analysis of the microstructural and property homogeneity of as-hot-pressed CoCr_13_Pt_16_B_10_ alloy ingot.

As described in Figure 1, a detailed assessment of the microstructure and property homogeneity within the as-hot-pressed billet was conducted with the exclusion of the hot top portion. To initiate this analysis, the microstructure and morphology of CoCr_13_Pt_16_B_10_ alloy after hot pressing were examined at fourteen distinct positions, denoted as $S_{1,1}^{xy}$ to $S_{1,7}^{xy}$ (the row along the margin) and $S_{2,1}^{xy}$ to $S_{2,7}^{xy}\;$(the row along the middle) along the cross-section. These positions spanned the width, starting from the side and progressing toward the central region, with seven samples examined for each row. Simultaneously, the microstructure and morphology at locations $S_{1,1}^{ij}\;$to $S_{1,7}^{ij}$ (the column along the margin) and $S_{2,1}^{ij}$ to $S_{2,7}^{ij}\;$(the column along the middle) on the longitudinal surface, extending from the top to the bottom, were also scrutinized. This comprehensive examination encompassed an evaluation of the microstructure and property uniformity throughout the cross-sectional and longitudinal aspects of the billet. The hardness and magnetic sampling methods are also shown in Figure 1.

### 3.2. Microstructure Uniformity

Figure 4 and Figure 5 present optical micrographs obtained from various sampling locations positioned in the proximity of the hot top area along the width of the CoCr_13_Pt_16_B_10_ alloy on the cross-sectional plane following hot pressing. Figure 4 shows the optical micrographs, the SDAS and fraction of dendrite phase variation of different sampling locations on the margin of the cross-section after hot pressing for CoCr_13_Pt_16_B_10_ alloy in Figure 1. Figure 5 shows the same characteristics in the middle of the cross-section for CoCr_13_Pt_16_B_10_ alloy in Figure 1.

It is evident on the comprehensive view of Figure 4 and Figure 5 that the dendrites at positions $S_{1,1}^{xy}$ and $S_{2,1}^{xy}$ are finer and the SDAS are smaller, while at positions $S_{1,7}^{xy}$ and $S_{2,7}^{xy}$, the dendrites are coarser and the SDAS are larger. The size and SDAS of the dendritic structure at positions $S_{1,1}^{xy}$ to $S_{1,7}^{xy}$ and $S_{2,1}^{xy}$ to $S_{2,7}^{xy}$ show an increasing trend; from the principles of solidification, it can be inferred that during the melting and casting processes, positions $S_{1,1}^{xy}$ and $S_{2,1}^{xy}$ are close to the mold wall and their heat diffuses rapidly outward, promoting a corresponding faster cooling rate. In addition, the mold wall can serve as the substrate for heterogeneous nucleation, forming more crystal nuclei and rapidly growing, which can readily form small micro-structures, known as surface fine grain areas in the solidification structure. At positions $S_{1,7}^{xy}$ and $S_{2,7}^{xy}$, due to the longer distance of the molten liquid front from the mold wall, the heat dissipation is slow, resulting in the formation of coarse dendritic structures. Simultaneously comparing the microstructure of the same row from $S_{1,1}^{xy}$ to $S_{1,7}^{xy}$ and $S_{2,1}^{xy}$ to $S_{2,7}^{xy}$ along the width, it was also found that the dendrite size and SDAS of the center layer through thickness namely, sample $S_{2,1}^{xy}$, with an analytical mean value of 9.63 μm increases to sample$S_{2,7}^{xy}$ with a mean value of 18.52 μm and are coarser than the outside layer, namely, sample $S_{1,1}^{xy}$ to $S_{1,7}^{xy}$, with an SDAS of 8.51 μm to 17.23 μm. The larger dendrite size and SDAS value of samples from $S_{2,1}^{xy}$ to $S_{2,7}^{xy}$ is also due to the comparatively long distance to samples from $S_{1,1}^{xy}$ to $S_{1,7}^{xy}$, and slower heat dissipation in the center layer of the ingot during solidification, making the probability of heterogeneous nucleation low, resulting in the formation of a coarser dendritic structure. Hence, it is discernible that the principal factor contributing to variations in dendrite size and SDAS values, both along the width and through the thickness direction, is the distinct cooling rates experienced during the casting process. It is noteworthy that, even after the hot pressing procedure, eradicating microstructural heterogeneity along the width direction remains an ongoing challenge.

Figure 4h and Figure 5h illustrate the SDAS and fraction of the dendrite arm area (mainly α-Co) for distinct sampling positions along the width direction in the proximity of the hot top area of theCoCr_13_Pt_16_B_10_ alloy after the hot pressing process. These figures distinctly reveal that the SDAS exhibits a gradual increase from the ingot’s edge toward its center. At the same time, there is a declining trend in the area occupied by the dendrite core phase. From the fitting results, it can be seen that both samples from the margin and middle area of the billet satisfy a relationship of parabolic law: $y=ax^{2}+bx+c$, with a = −0.19, b = 2.93, and c = 6.03 for samples from $S_{1,1}^{xy}$ to $S_{1,7}^{xy}$ and a= −0.25, b = 3.49, and c = 6.38 for samples from $S_{2,1}^{xy}$ to $S_{2,7}^{xy}$. For the dendrite arm phase fraction, a line relationship y=mx+n was obtained with m = −2.31 and n = 89.38 for the margin region samples and m = −2.99 and n = 89.20 for the samples in the middle.

Figure 6 and Figure 7 show microstructure photographs of the longitudinal surfaces of different sampling locations along the height direction on the surface for CoCr_13_Pt_16_B_10_ alloy after hot pressing.

From Figure 6 and Figure 7, it can be seen that the relationship of SDAS and the dendrite arm area fraction with sampling spots shows nearly the same trend for different columns, but the relative value of which varies from $S_{1,1}^{ij}$ to $S_{1,7}^{ij}$ and $S_{2,1}^{ij}$ to $S_{2,7}^{ij}$. The positions from $S_{2,1}^{ij}$ to $S_{2,7}^{ij}$ are closer to the middle of the surface compared with $S_{1,1}^{ij}$ to $S_{1,7}^{ij}$, with a larger SDAS and smaller dendrite phase area proportion. The observed variation in values can be attributed to differences in solidification rates, where areas closer to the center of the surface experience slower solidification, leading to a coarser microstructure. Conversely, as shown in Figure 6h and Figure 7h, a uniform trend in SDAS and the dendrite arm phase faction occurs along the height direction due to the equidistant proximity from the mold wall to the sampling locations, resulting in a consistent cooling rate. Consequently, the SDAS and the fraction of the dendrite arm area exhibit minimal variation along the height direction. In a parallel manner, the inherent microstructural heterogeneity persists on the longitudinal surface even after the hot pressing process.

### 3.3. Vickers Hardness

Table 3 provides a comprehensive overview of the Vickers hardness values for the CoCr_13_Pt_16_B_10_ alloy before and after the hot pressing procedure.

To assess the disparities in hardness, hardness assessments were conducted on the dendrite arm and interdendritic region. For each sample near the hot top area, three distinct spots within each region were meticulously selected for analysis.

The Vickers hardness value is consistently lower in the dendrite core than in the interdendritic area for the as-cast and as-hot-pressed samples. This disparity can be attributed to element segregation and the inherent strengthening mechanisms within the alloy. Comparatively, following the hot pressing treatment, a reduction in hardness is observed in contrast to the values exhibited by the as-cast ingot alloy. Simultaneously, a marked enhancement in uniformity is noted. The hardness values in the dendrite arm area decrease by approximately 7%, while those in the interdendritic area exhibit a reduction of nearly 3%. Consequently, the hot pressing process contributes to a reduction in hardness, with a more pronounced effect on the dendrite arm region. It is evident that hot pressing significantly improves the deformation performance of the CoCr_13_Pt_16_B_10_ alloy.

Figure 8 shows the Vickers hardness of the CoCr_13_Pt_16_B_10_ alloy for various positions along the cross-section, encompassing the dendrite core and interdendritic region, subsequent to the hot pressing process.

It can be seen from Figure 8 that the hardness of the interdendritic region on the cross-section has a similar trend from positions $S_{1,1}^{xy}$ to $S_{1,7}^{xy}$ and $S_{2,1}^{xy}$ to $S_{2,7}^{xy}$, with only slight variation, while the hardness of the dendrite arm area shows a decreasing trend. The change in hardness is closely related to the change in microstructure structure. At positions $S_{1,1}^{xy}$ and $S_{2,1}^{xy}$, the microstructure is finer; the finer the structure, the more grain boundaries. This hinders the movement of dislocations, increases the deformation resistance of the material, and increases its strength, resulting in a higher hardness value. At positions $S_{1,7}^{xy}$ and $S_{2,7}^{xy}$ the microstructure is coarse with fewer grain boundaries, the deformation resistance of the material is smaller, and its strength is lower, resulting in a correspondingly lower hardness value. For CoCr_13_Pt_16_B_10_ alloy, its dendrite structure is well-developed, and the SDAS is easy to measure. However, the complete grain boundaries are difficult to observe.

According to the Hall Petch relationship [22,23], the relationship between grain size and alloy strength can be expressed as
(2)$$\sigma_{s}=\sigma_{o}+{Kd^{-}}^{\frac{1}{2}}$$
where $\sigma_{s}$ is the yield strength, $\sigma_{o}$ represents the frictional stress that hinders the movement of dislocations within the grain, K represents the influence of grain boundaries on deformation, which is related to the grain boundary structure, and d represents the grain size or the interlayer spacing between two-phase structures. From Equation (2), it can be seen that as the grain size increases, the strength of the alloy decreases. The empirical relationship between the yield strength and hardness of the alloy satisfies Equation (3) [22]:(3)$$H_{v}\approx 3\sigma_{s}$$

Hence, it can be deduced that the correlation between the hardness and grain size of the alloy adheres to the Hall-Petch relationship, where an increase in grain size within the alloy leads to a corresponding decrease in hardness. This relationship between hardness and grain size can be established by amalgamating Equations (2) and (3) into a unified expression, denoted as Equation (4):
(4)$$H_{v}={Hv}_{o}+k_{1}d^{{-}_{2}^{1}}$$

Therefore, this paper uses the SDAS (λ_2_) to replace the grain size (*d*) and bring it into Equation (4) to analyze the variation of SDAS on hardness. This can be expressed as Equation (5):(5)$$H_{v}={Hv}_{o}+k_{1}\lambda_{2}^{{-}_{2}^{1}}$$

The relationship between the reciprocal square root of SDAS and Vickers hardness is shown in Figure 9.

The experimental data does not agree well with the Hall-Petch relationship. The Vickers hardness of the interdendritic area shows greater dispersion. The variation in the calculated correlation constants ${Hv}_{o}$ and $k_{1}$ can be attributed to the nature of the secondary dendrite spacing, which represents a statistical average of measured values. For the dendrite arm region, both primary and secondary dendrites exist. Due to differences in equilibrium distribution coefficients, solute segregation would occur, resulting in the concentration difference between the primary and secondary dendrite areas and, in turn, the hardness variation. Meanwhile, for the interdendritic area, high-order dendrites and segregation coexist, enhancing the inhomogeneity of the composition and the hardness. Additionally, Equation (3) may not be readily maintained in the CoCrPtB alloy [22]. As a result, fluctuations in the disparity of SDAS and hardness are observed.

Figure 10 shows the Vickers hardness values for the CoCr_13_Pt_16_B_10_ alloy at various positions along the longitudinal surface, encompassing the dendrite core and interdendritic region, after the hot pressing process.

From positions $S_{1,1}^{ij}$ to $S_{1,7}^{ij}$ and $S_{2,1}^{ij}$ to $S_{2,7}^{ij}$, the hardness of the dendrite arm and interdendritic structures tends to agree for the same column. In contrast, the hardness at $S_{2,1}^{ij}$ to $S_{2,7}^{ij}$ is lower than that at $S_{1,1}^{ij}$ to $S_{1,7}^{ij}$. This is because the microstructure from $S_{1,1}^{ij}$ to $S_{1,7}^{ij}$ and $S_{2,1}^{ij}$ to $S_{2,7}^{ij}$ is consistent, and the microstructure of the column at positions $S_{1,1}^{ij}$ to $S_{1,7}^{ij}$ is finer than from $S_{2,1}^{ij}$ to $S_{2,7}^{ij}$; thus, the hardness at positions $S_{1,1}^{ij}$ to $S_{1,7}^{ij}$ is higher than that from$S_{2,1}^{ij}$ to $S_{2,7}^{ij}$.

### 3.4. Magnetic Characteristics

Figure 11 illustrates the hysteresis loops for the as-cast and as-hot-pressed CoCr_13_Pt_16_B_10_ alloy.

Those loops enable the derivation of relevant magnetic characteristic parameters, also summarized in Figure 11. The coercivity of the as-cast sample is 2416 A/m, and the remanent magnetism B_r_ and maximum magnetic energy product (BH)_max_ are 0.047 T and 0.021 J/m^3^, respectively, with a squareness ratio, R_s_, of 0.29. After hot pressing, the magnetic performance parameter values decreased. The coercivity value decreased significantly by an amplitude of 526 A/m. Coercivity is a structurally sensitive magnetic parameter that can be significantly altered by grain size, grain arrangement, orientation, machining (such as deformation), and heat treatment.

The magnetic performance parameters exhibit a decline after hot pressing, signifying a reduction in the magnetic performance of the alloy due to microstructural alterations incurred during the hot pressing process. The X-ray diffraction (XRD) analysis revealed that the CoCr_13_Pt_16_B_10_ alloy comprises a face-centered cubic (fcc) structure α-Co phase and a hexagonal close-packed (hcp) structure ε-Co phase. For the ε-Co phase, [0001] direction is the easily magnetized axis with strong magnetic crystal anisotropy. The magnetic properties of the alloy are mainly composed of hcp structures provided by the ε-Co phase. From XRD results, it can be inferred that after hot pressing, the ε-Co phase content is decreased, resulting in decreased magnetic properties of CoCr_13_Pt_16_B_10_ alloy.

The magnetic parameters of different sampling locations for the CoCr_13_Pt_16_B_10_ alloy after hot pressing are listed in Table 4.

Note that samples $S_{A,E}^{ij}$, $S_{B,E}^{ij}$, and $S_{C,E}^{ij}$ are nearly at the same position as samples $S_{1,2}^{xy}$, $S_{1,4}^{xy}$, and $S_{1,6}^{xy}$; samples $S_{A,I}^{ij}$, $S_{B,I}^{ij}$, and $S_{C,I}^{ij}$ are nearly at the same position as $S_{2,2}^{xy}$, $S_{2,4}^{xy}$, and $S_{2,6}^{xy}$ as measuring the SDAS shown in Figure 4. It is apparent that, for the same row, from samples $S_{A,E}^{ij}$ to $S_{C,E}^{ij}$ and $S_{A,I}^{ij}$ to $S_{C,I}^{ij}$, the B_r_, (BH)_max_ and R_s_ are relatively the same with the coercivity decreasing along the width direction on the cross-section. The relationship between the SDAS and magnetic parameters of the CoCr_13_Pt_16_B_10_ alloy after hot pressing is shown in Figure 12.

When transitioning from samples denoted as $S_{A,E}^{ij}$ to $S_{C,E}^{ij}$, and $S_{A,I}^{ij}$ to $S_{C,I}^{ij}$ an evident pattern emerges. As the mean SDAS increases, the magnetic parameters, including B_r_, (BH)_max_, and R_s_, exhibit minimal variation, remaining relatively consistent. In contrast, coercivity demonstrates a marked reduction. This observation underscores the heightened sensitivity of coercivity to microstructural alterations compared to other magnetic parameters. The simultaneous augmentation of SDAS and ε-Co phase content across different sample positions along the width direction stands out as the underlying cause for the observed decline in coercivity.

Meanwhile, for samples along the longitudinal section, namely $S_{D,E}^{ij}$, $S_{E,E}^{ij}$, and $S_{F,E}^{ij}$ in the exterior regions and $S_{D,I}^{ij}$, $S_{E,I}^{ij}$, and $S_{F,I}^{ij}$ within the interior regions of the billet, the magnetic properties are shown at Table 4. For the same longitudinal layers, the magnetic properties from $S_{D,E}^{ij}$ to $S_{F,E}^{ij}$ or $S_{D,I}^{ij}$ to $S_{F,I}^{ij}$ exhibit relatively no change, respectively. However, the coercivity of samples from $S_{D,I}^{ij}$ to $S_{F,I}^{ij}$ is lower than the other longitudinal layer from $S_{D,E}^{ij}$ to $S_{F,E}^{ij}$. In a similar vein, it can be deduced that the SDAS has a discernible influence, leading to variations in the dendrite fraction from the exterior to the interior region.

## 4. Conclusions

The as-cast CoCr_13_Pt_16_B_10_ alloy displays a characteristic dendritic microstructure, primarily comprising a face-centered cubic (fcc) α-Co phase and a minor hexagonal close-packed (hcp) ε-Co phase. Notably, a discernible negative segregation of Pt is observed within the dendrite arms, while Cr exhibits negligible segregation.The microstructure of the billet after the hot pressing process exhibits an inherent inhomogeneity. Notably, the following observations were made: (a) In the vicinity of the uppermost section on the cross-sectional plane, there is a discernible increasing trend in dendritic structure size and SDAS along the width direction. Moreover, upon making row-to-row comparisons, it is evident that samples positioned within the central layer, spanning the thickness of the billet, manifest a coarser dendritic structure size and larger SDAS. (b) Conversely, when examining the samples taken from the longitudinal surface, running along the height dimension, there is relatively minimal variation observed in the SDAS and dendrite arm area fraction.The Vickers hardness values in the dendrite core are consistently lower than those observed in the interdendritic region for the as-cast and as-hot-pressed samples. After the hot pressing procedure, a reduction in Vickers hardness is noticeable, with the hardness in the dendrite arm area decreasing by approximately 7%, and in the interdendritic area by nearly 3%. The experimental findings on the cross-sectional plane are not closely aligned with the Hall-Petch relationship, demonstrating an inhomogeneity correlation between hardness and SDAS. However, no noteworthy differences are observed near the longitudinal surface, where variations in hardness are relatively minimal.The initial coercivity of the as-cast sample is 2416 A/m, accompanied by respective values of 0.047 T for remanence (B_r_), 0.021 J/m^3^ for maximum energy product ((BH)_max_), and a relative squareness (R_s_) coefficient of 0.29. However, after hot pressing treatment, a substantial decline is observed in the magnetic performance parameters, particularly coercivity, which experiences a noteworthy reduction. Following hot pressing, the parameters B_r_, (BH)_max_, and R_s_ exhibit minimal change, while coercivity experiences a significant decrease, specifically along the width on the cross-sectional plane. In contrast, when considering the samples longitudinally, the coercivity values consistently remain lower than those within the same column.The microstructure, Vickers hardness, and magnetic properties of the CoCr_13_Pt_16_B_10_ alloy demonstrate nonhomogeneity after applying vacuum hot pressing. Nevertheless, the solidification rate plays a significant role in generating microstructural variations. Notably, variations in SDAS and the relative ratios of distinct Co phases primarily account for the observed discrepancies in Vickers hardness and magnetic properties.In the future, more work is needed, including more precise control and monitoring of cooling rates, thorough statistical analysis with in-depth characterization techniques, application of different alloy compositions, etc.

## Figures and Tables

**Figure 1 materials-17-00544-f001:**
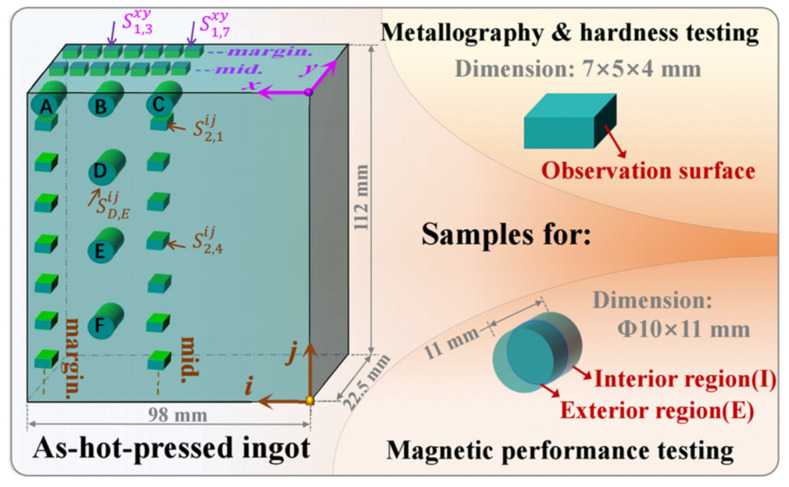
Schematic diagram of sampling for CoCr_13_Pt_16_B_10_ alloy after hot pressing.

**Figure 2 materials-17-00544-f002:**
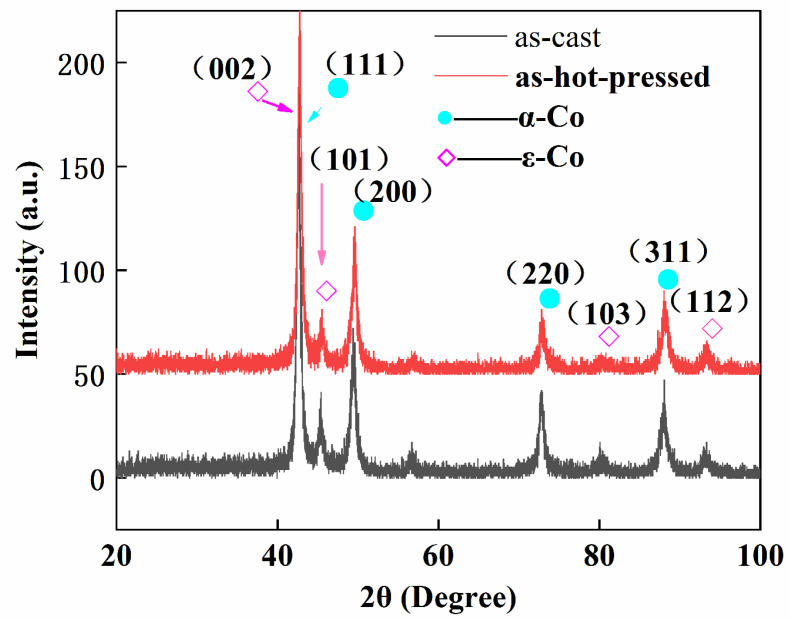
XRD patterns of CoCr_13_Pt_16_B_10_ alloys for as-cast and as-hot-pressed samples.

**Figure 3 materials-17-00544-f003:**
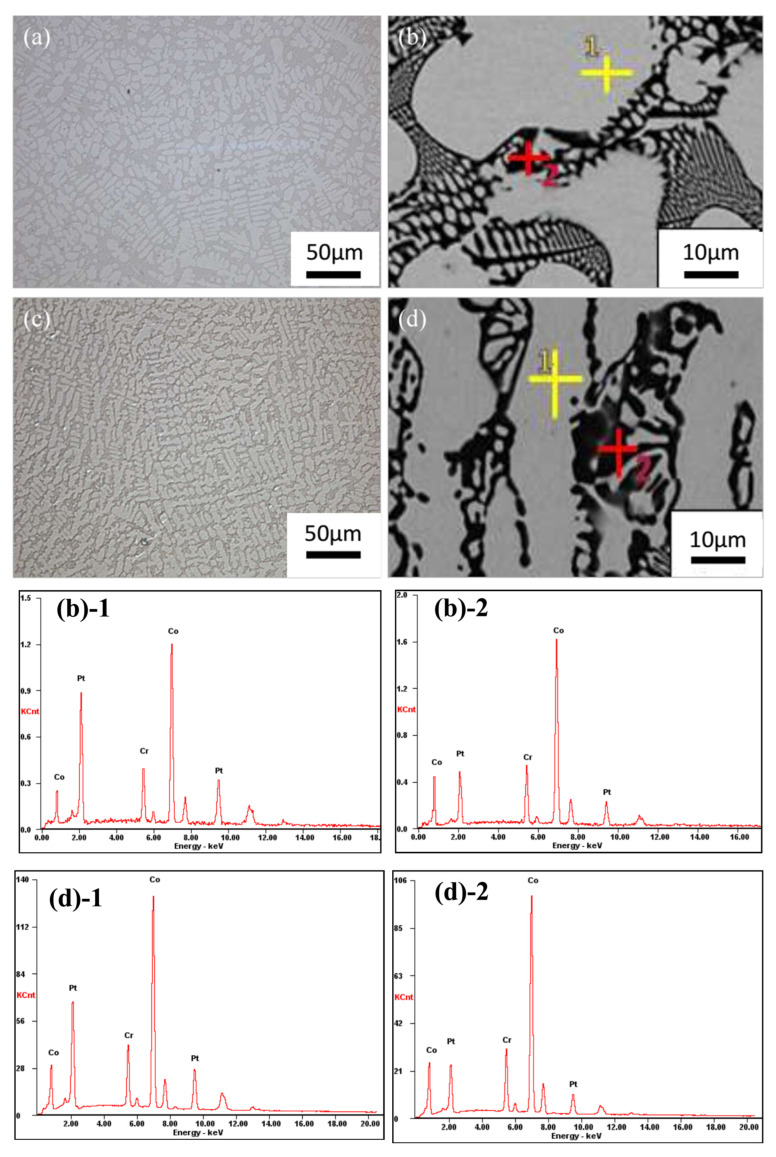
OM and SEM micrographs of CoCr13Pt16B10 alloys (**a**) OM image of the as-cast specimen and (**b**) SEM and EDS analysis spots for different areas of the as-cast sample, where point 1 is the dendrite arm and point 2 shows the dark region in the inter-dendrite region, (**b**)**-1** is the EDS of point 1 in (**b**), (**b**)**-2** is the EDS of point 2 in (**b**); (**c**) OM image of the as-hot-pressed specimen; (**d**) SEM and EDS analysis spots for different areas of the as-hot-pressed sample, where point 1 denotes the dendrite arm and point 2 the dark area in the inter-dendrite region, (**d**)**-1** is the EDS of point 1 in (**d**), (**d**)**-2** is the EDS of point 2 in (**d**).

**Figure 4 materials-17-00544-f004:**
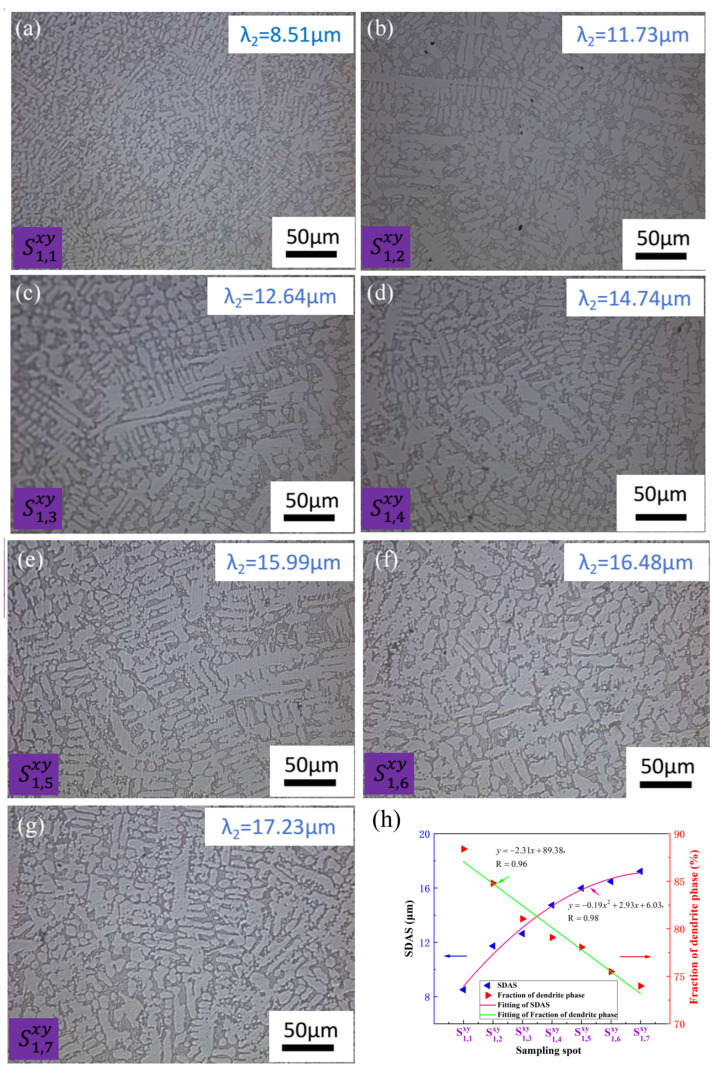
The optical micrographs of different sampling locations on the margin of the cross-section after hot pressing the CoCr_13_Pt_16_B_10_ alloy in Figure 1: (**a**) $S_{1,1}^{xy}$, (**b**)$\;S_{1,2}^{xy}$, (**c**) $S_{1,3}^{xy}$, (**d**) $S_{1,4}^{xy}$, (**e**) $S_{1,5}^{xy}$, (**f**) $S_{1,6}^{xy};$ and (**g**) $S_{1,7}^{xy}$, from the edge to center; (**h**) SDAS and fraction of dendrite phase variation with sampling spot.

**Figure 5 materials-17-00544-f005:**
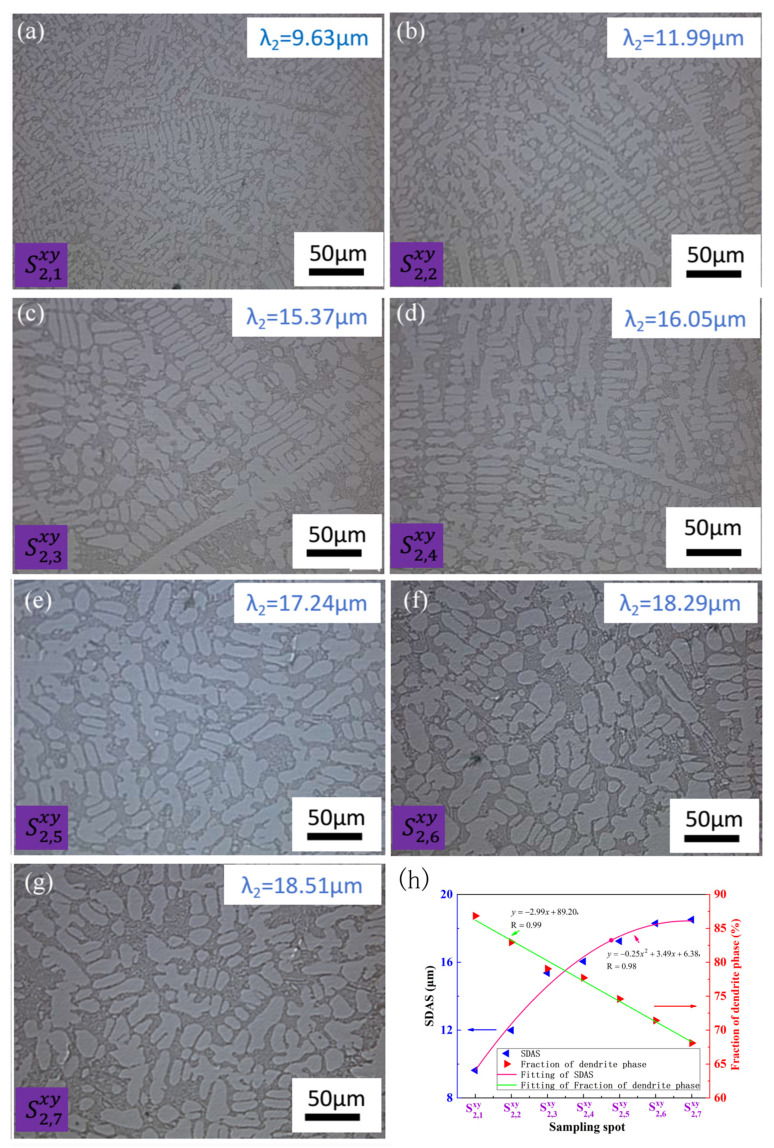
The optical micrographs of different sampling locations in the middle of the cross-section after hot pressing for CoCr_13_Pt_16_B_10_ alloy in Figure 1: (**a**) $S_{2,1}^{xy}$, (**b**) $S_{2,2}^{xy}$, (**c**) $S_{2,3}^{xy}$, (**d**) $S_{2,4}^{xy}$, (**e**) $S_{2,5}^{xy}$, (**f**) $S_{2,6}^{xy}$, and (**g**) $S_{2,7}^{xy}$, from edge to the center; (**h**) SDAS and fraction of dendrite phase variation with sampling spot.

**Figure 6 materials-17-00544-f006:**
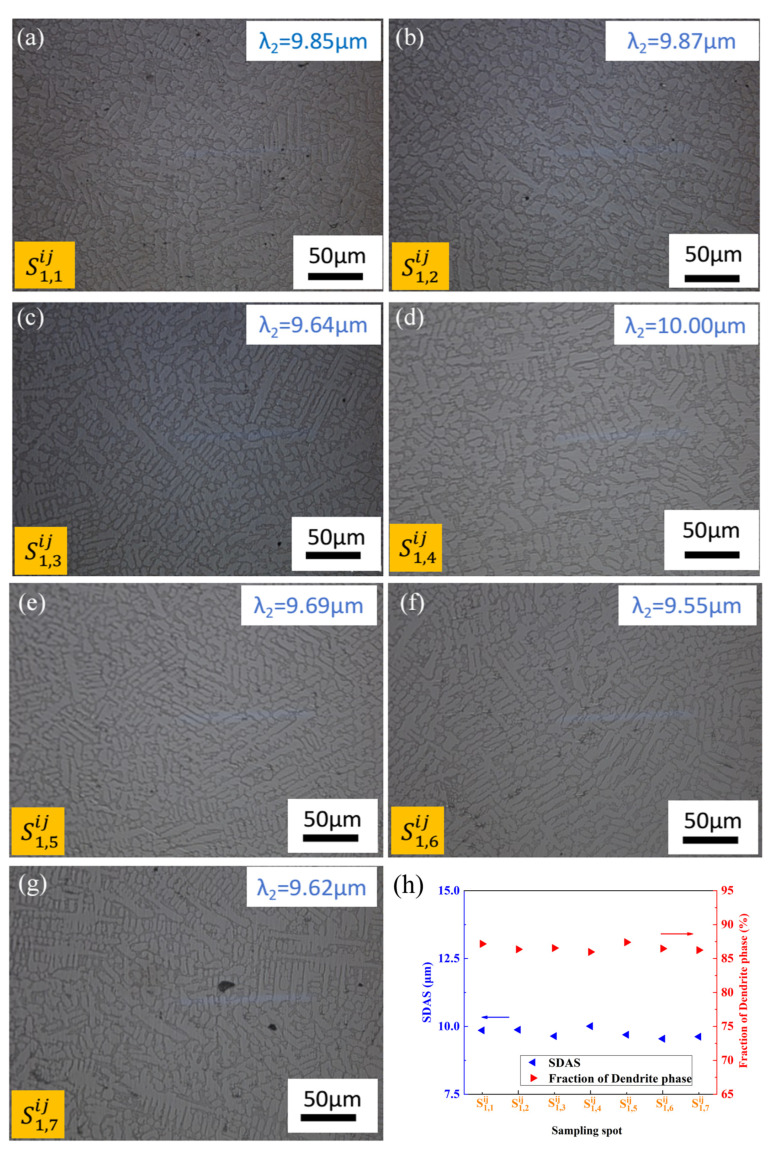
Optical micrographs of the longitudinal section for different sampling locations along the height direction after hot pressing for CoCr_13_Pt_16_B_10_ alloy in Figure 1 on the margin: (**a**) $S_{1,1}^{ij}$, (**b**) $S_{1,2}^{ij}$, (**c**)$\;S_{1,3}^{ij}$, (**d**) $S_{1,4}^{ij}$, (**e**)$\;S_{1,5}^{ij}$, (**f**)$\;S_{1,6}^{ij}$, and (**g**) $S_{1,7}^{ij}$ from edge to the center. (**h**) SDAS and fraction of dendrite phase variation with sampling spot.

**Figure 7 materials-17-00544-f007:**
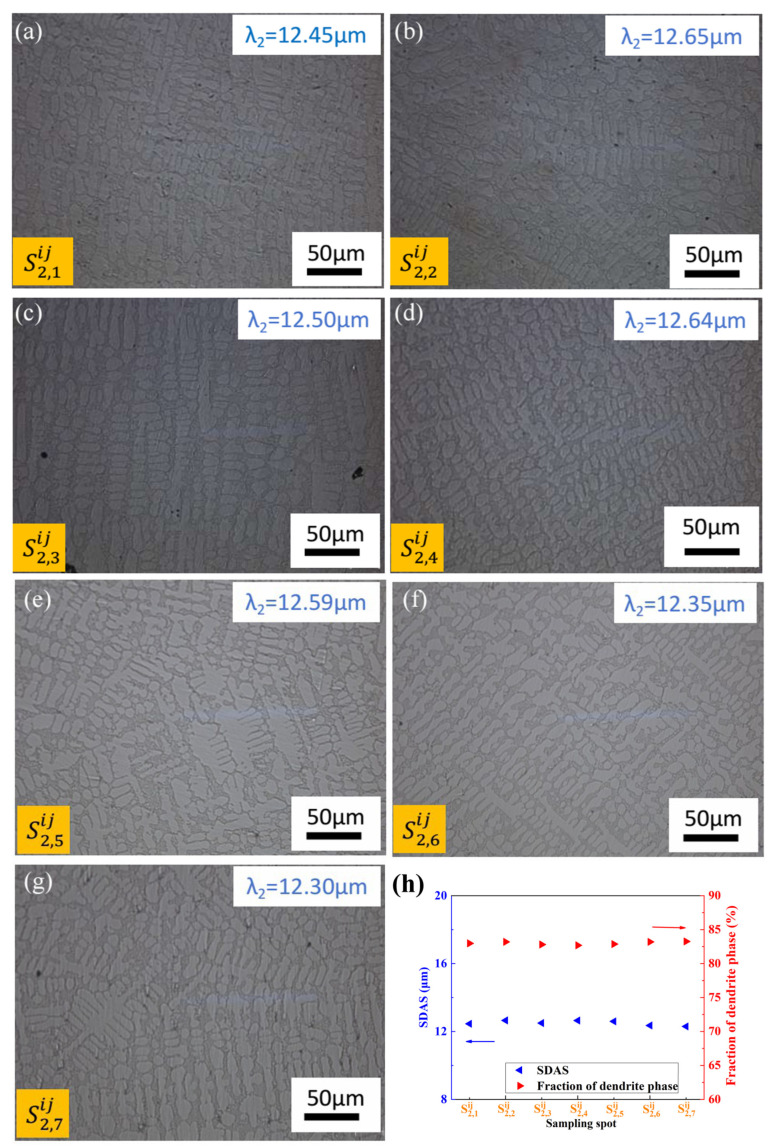
Optical micrographs of the longitudinal 
section for different sampling locations along the height direction after hot 
pressing for the CoCr_13_Pt_16_B_10_ alloy in Figure 1 in the middle: (**a**)${\;S}_{2,1}^{ij}$, (**b**)$\;S_{2,2}^{ij}$, (**c**) $S_{2,3}^{ij}$, (**d**) $S_{2,4}^{ij}$, (**e**)$\;S_{2,5}^{ij}$, (**f**) $S_{2,6}^{ij}$, and (**g**) $S_{2,7}^{ij}$ from the edge to the center. (**h**) SDAS and fraction of dendrite phase variation with sampling spot.

**Figure 8 materials-17-00544-f008:**
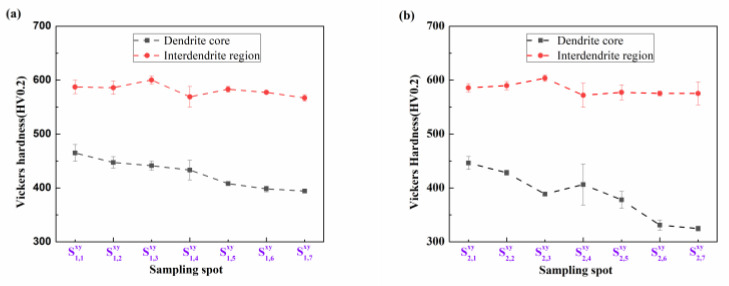
Vickers hardness of CoCr_13_Pt_16_B_10_ alloy for different locations along the cross-section for the dendrite core and interdendritic region after hot pressing. (**a**) Margin area from sample ${\;S}_{1,1}^{xy}$ to ${\;S}_{1,7}^{xy}$ and (**b**) the middle area from sample ${\;S}_{2,1}^{xy}$ to ${\;S}_{2,7}^{xy}$.

**Figure 9 materials-17-00544-f009:**
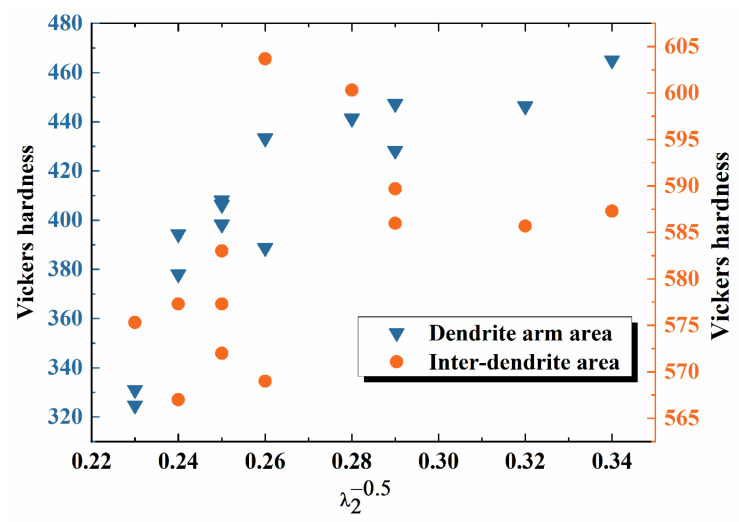
Variation in hardness with the reciprocal square root of mean SDAS ($\lambda_{2}^{-0.5}$) of the dendrite arm area and the interdendritic area of the CoCr_13_Pt_16_B_10_ alloy after hot pressing.

**Figure 10 materials-17-00544-f010:**
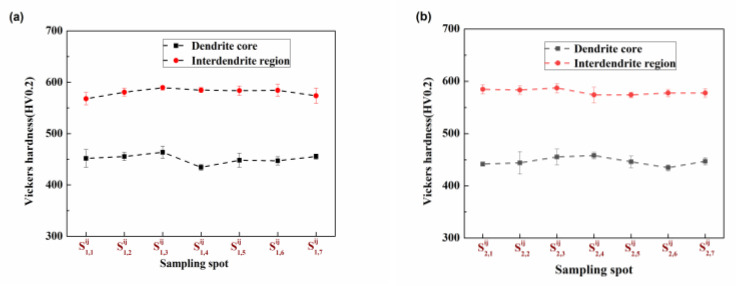
Vickers hardness of the CoCr_13_Pt_16_B_10_ alloy for different locations on the longitudinal surface for the dendrite core and interdendritic region after hot pressing. (**a**) Samples $S_{1,1}^{ij}$ to $S_{1,7}^{ij}$ and (**b**) $S_{2,1}^{ij}$ to $S_{2,7}^{ij}$.

**Figure 11 materials-17-00544-f011:**
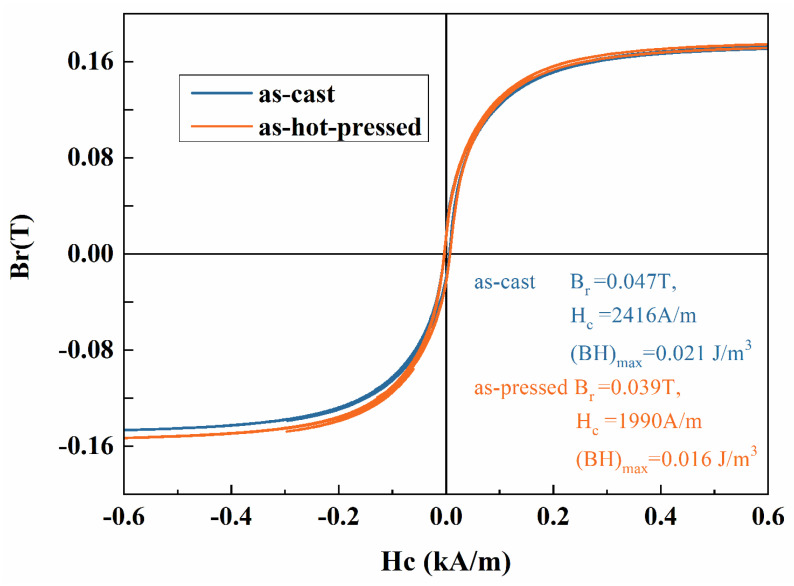
Hysteresis loop of the CoCr_13_Pt_16_B_10_ alloy for as cast and as-hot-pressed samples.

**Figure 12 materials-17-00544-f012:**
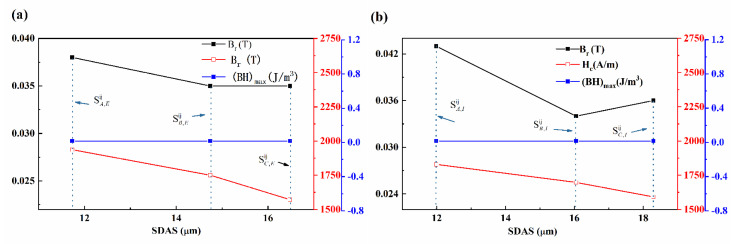
Relationship between SDAS and magnetic parameters of the CoCr_13_Pt_16_B_10_ alloy after hot pressing process. (**a**) Samples$S_{A,E}^{ij}$, $S_{B,E}^{ij}$, and $S_{C,E}^{ij}$; (**b**) samples $S_{A,I}^{ij}$, $S_{B,I}^{ij}$, and $S_{C,I}^{ij}$.

**Table 1 materials-17-00544-t001:** Chemical composition analysis result of CoCr_13_Pt_16_B_10_ alloy (wt.%).

Element	Co	Cr	Pt	B
nominal	47.93	9.01	41.62	1.44
actual	47.90	8.96	41.73	1.41

**Table 2 materials-17-00544-t002:** Distribution of the major elements in the dendrite core and inter-dendrite region in as-cast and as-hot-pressed CoCr_13_Pt_16_B_10_ alloys (wt.%).

Sample	Element	Dendrite Core	Inter-Dendrite Region	k
as-cast	Cr	9.97	10.12	0.99
	Pt	52.34	16.30	3.21
as-hot-pressed	Cr	9.77	9.52	1.02
	Pt	49.29	15.25	3.23

**Table 3 materials-17-00544-t003:** Vickers hardness (Hv_0.2_) of CoCr_13_Pt_16_B_10_ alloys for as-cast and as-hot-pressed samples (kg/mm^2^).

Sample	Testing Area	HV-1	HV-2	HV-3	HV (Average Value)
as-cast	dendrite core	494	485	477	485
	interdendrite region	618	630	623	624
as-hot-pressed	dendrite core	444	447	453	448
	interdendrite region	592	596	587	592

**Table 4 materials-17-00544-t004:** Magnetic parameters of different sampling locations for the CoCr_13_Pt_16_B_10_ alloy after hot pressing (position of samples can be seen in Figure 1).

Sampling Position	Sample	B_r_(T)	H_c_(A/m)	(BH)_max_(J/m^3^)	R_s_
cross section	$S_{A,E}^{ij}$	0.038	1936.7	0.016	0.24
	$S_{B,E}^{ij}$	0.035	1751.0	0.016	0.24
	$S_{C,E}^{ij}$	0.035	1572.0	0.016	0.24
	$S_{A,I}^{ij}$	0.043	1830.0	0.016	0.24
	$S_{B,I}^{ij}$	0.034	1697.7	0.016	0.23
	$S_{C,I}^{ij}$	0.036	1592.0	0.016	0.24
longitudinal surface	$S_{D,E}^{ij}$	0.043	1983.7	0.017	0.23
	$S_{E,E}^{ij}$	0.039	1990.0	0.016	0.24
	$S_{F,E}^{ij}$	0.042	1990.3	0.016	0.24
	$S_{D,I}^{ij}$	0.046	1856.7	0.016	0.23
	$S_{E,I}^{ij}$	0.035	1830.0	0.016	0.24
	$S_{F,I}^{ij}$	0.040	1803.7	0.017	0.23

## Data Availability

The authors confirm that the data supporting the findings of this study are available within the article. Data are available on request due to restrictions on the Yunnan precious Metal Lab.

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
