# Peer review of "Microstructural Heterogeneity and Property Variations in Cast and Vacuum Hot-Pressed CoCrPtB Alloy"

_materials, 2024, doi:10.3390/ma17030544_

Round 1
Reviewer 1 Report
Comments and Suggestions for Authors
The authors have presented an investigation for exploring the Secondary Dendritic Arm Spacing (SDAS), Vickers hardness, and magnetic characteristics of a CoCrPtB alloy, which is synthesized through the casting and hot-pressing processes. The work reflects an incremental but significant contribution to the field of materials' analysis for special purpose alloys. However, there are notable scientific and/or technical inconsistencies, as mentioned below, in the reported work, which must be addressed adequately in order to be considered it for further review.
1. In materials and methods section, the authors have explained the formation of alloy. Is this a standard procedure or a novel? If it is, it must be stated and cited. If not, then the authors are required to include few more details. For example, in last step, the molten alloy is cooled to room temperature. In this case, what is the cooling rate? Is it just poured in the bucket and let it cool at an open ambient conditions? Since all these scenarios will greatly affect the properties of an alloy. It is required to include all these details with the cooling curve if available.
2. It is also required to include the details about specimen preparation steps like polishing operation, as in microstructure observation section. Furthermore, what kind of chemical treatment was performed on specimen before optical and electron micrography?
3. It is recommended to include all EDS graphs for Figure 3.
4. It is required to discuss Figure 4 in detail.
5. The authors are also recommended to include the possible future developments in the reported work OR how someone could build further by using the reported work as a benchmark.
Comments on the Quality of English LanguageIt is highly recommended to perform proofreading for any typo or grammatical errors.
Author Response
1. In materials and methods section, the authors have explained the formation of alloy. Is this a standard procedure or a novel? If it is, it must be stated and cited. If not, then the authors are required to include few more details. For example, in last step, the molten alloy is cooled to room temperature. In this case, what is the cooling rate? Is it just poured in the bucket and let it cool at an open ambient conditions? Since all these scenarios will greatly affect the properties of an alloy. It is required to include all these details with the cooling curve if available.
A:The formation of alloy is nearly a standard procedure. Normally,to forming an alloy ingot, one must mixing different alloy elements together and melt their into a ingot. The difference between manufacture methods is the specific parameter utilized, such as whether use master alloy, the temperature or pouring time. In this paper, the mold used for pouring is graphite mold, after pouring, the ingot was left in the chamber to cooling down, and take out for 1 hour later. This was added in the revised manuscript.
2. It is also required to include the details about specimen preparation steps like polishing operation, as in microstructure observation section. Furthermore, what kind of chemical treatment was performed on specimen before optical and electron micrography?
A: All the specimens are prepared by manual grinding with Al2O3 paper followed by polishing with diamond abrasives with the size of 15μm、9μm、6μm successively. Finally, the excess abrasive particles were removed from the polished surfaces using an ultrasonic cleaner with acetone and water.This was added in the revised manuscript.
3. It is recommended to include all EDS graphs for Figure 3.
A: All EDS graphs was included in Figure3.
4.It is required to discuss Figure 4 in detail.
A: The descriptive content of Figure 4 was added “Figure 4 shows the optical micrographs, the SDAS and fraction of dendrite phase variation of different sampling locations on the cross-section after hot pressing for CoCr13Pt16B10 alloy in Figure 1 on the margin.while Figure 5 shows the same characteristics for CoCr13Pt16B10 alloy in Figure 1 in the middle.”
5.The authors are also recommended to include the possible future developments in the reported work OR how someone could build further by using the reported work as a benchmark.
A: Some future works are proposed at the end of the paper.
Reviewer 2 Report
Comments and Suggestions for Authors
Dear Authors:
Remarks:
1. The introduction should be revised. The sentence “Consequently, there exists a well-established body of research dedicated to the exploration of the relationship between dendritic structures and the properties of materials [1-11].” should be divided and particular references should be discussed separately. A significant part of the introduction is related to the description of the influence of the chemical components on the properties, whereas, the main scope of the paper is only mentioned. It should be improved.
2. Line 70 „The alloy composition employed in this study consists of CoCr13Pt16B10”. – Please check and verify. CoCr13Pt16B10 looks like a designation of the material.
3. Figure 1 is blurred. I propose to use more contrasting colors instead of yellow color.
4. Line 104 “45KV" – Please check and verify.
5. Lines 104 -106 “The XRD data was acquired at a scanning rate of 5 /min, with a step size of 0.02, encompassing a 2θ range spanning from 20 to 100” – Please check units and verify. What is θ?
6. Line 111 “An average value of at least 100 measures [1]” – Reference is not clear.
7. Line 119 “The chemical composition analysis of CoCr13Pt16B10 alloy is listed in Table 1” and line 122 “Figure 2 is the XRD patterns “ – grammar/style.
8. Almost all figures are blurred and should be improved. In many cases, it is difficult to read descriptions.
9. Line 144 “As depicted in Figure 3a and Figure 3b are the OM and SEM micro-graphs” – grammar
10. Lines 168-172 – what in the investigated case means positive/negative segregation?
11. Lines 286 -288 “From Figure 6 and Figure 7, it can be seen that the relation of SDAS and the fraction of dendrite arm area with sampling spots show nearly the same trend but the value of which is varied from S1ij,1 to S1ij,7 and S2ij,1 to S2ij,7.” – not clear.
12. Table 3 should be improved ( descriptions of the columns).
13. Equation 2 and line 339 “σs is the strength” – what strength, please precise.
14. Equation 3 – what is σ?
15. Equation 4 and Line 351 “Therefore, this paper uses the SDAS (λ2 ) to replace the grain size and bring it into equation (4) to analyze the variation of size of dendrite on hardness.” – Not clear. Please provide the mentioned equation or specify more precisely.
16. Line 357 “It can be seen that the experimental data is not in good agreement with the Hall-Petch relationship.” What is presented in Figure 9 – experimental or analytical results? Where is verification of the Hall-Petch relationship What is the basis for this statement?
Kind Regards,
Comments on the Quality of English LanguageThe grammar and style should be coprehensively improved. The more detailed comments are given in Comments and Suggestions for Authors.
Author Response
1. The introduction should be revised. The sentence “Consequently, there exists a well-established body of research dedicated to the exploration of the relationship between dendritic structures and the properties of materials [1-11].” should be divided and particular references should be discussed separately. A significant part of the introduction is related to the description of the influence of the chemical components on the properties, whereas, the main scope of the paper is only mentioned. It should be improved.
A: It was modified, more discriptions were added.
2. Line 70 „The alloy composition employed in this study consists of CoCr13Pt16B10”. – Please check and verify. CoCr13Pt16B10 looks like a designation of the material.
A: The theoretical composition of the alloy studied in this paper is Co-(13at.%)Cr-(16at.%)Pt-(10at.%)B. normally, Both Co-(13at.%)Cr-(16at.%)Pt-(10at.%)B or CoCr13Pt16B10 are or the latter was used and subscripts were applied and modified in the manuscript.
3. Figure 1 is blurred. I propose to use more contrasting colorsinstead of yellow color.
A: Already modified, coffee color was used instead of yellow color. But when considering the color-coding of different sampling position applied in this paper, yellow seems to be acceptable, the reason for the color is not contrasting enough is due to the low resolution. For the Figure’s blur issue please see answers to question 8.
4. Line 104 “45KV" – Please check and verify.
A: the voltage of XRD operation is 45KV, it was modified. So did the current.
5. Lines 104 -106 “The XRD data was acquired at a scanning rate of 5 /min, with a step size of 0.02, encompassing a 2θ range spanning from 20 to 100” – Please check units and verify. What is θ?
A: The unit degree was missing during the manuscript transforming process, it was added now. θ stands for the degree of diffraction angle.
6. Line 111 “An average value of at least 100 measures [1]” – Reference is not clear.
A: As shown in Fig.4 of reference[1], when 100 -values were measured at each position from the chillshell interface the influence of possible measurement errors on the final result can be minimized. This was added in the manuscript.
7. Line 119 “The chemical composition analysis of CoCr13Pt16B10 alloy is listed in Table 1” and line 122 “Figure 2 is the XRD patterns “ – grammar/style.
A: they were modified respectively as “The chemical composition analysis result of CoCr13Pt16B10 alloy is listed in Table 1” and “Figure 2 shows the XRD patterns”.
8. Almost all figures are blurred and should be improved. In many cases, it is difficult to read descriptions.
A: Actually, the original figures in the submitted manuscript is clear and prepared according to the requirements in “Instructions for Authors”, and high resolution figures were attached when submitted. We will contact the editors to fix this.
9. Line 144 “As depicted in Figure 3a and Figure 3b are the OM and SEM micro-graphs” – grammar
A:It was changed to “Figure 3 shows the OM and SEM micro-graphs of as-cast and as-hot-pressed CoCr13Pt16B10 samples”.
10. Lines 168-172 – what in the investigated case means positive/negative segregation?
A: Line 170 ,the as-hot-press Pt column was missing, and it was added again. As described in the paper, K is the result of relatively elements content of the dendrite core and inter-dendrite region.When <1, the element exhibits positive segregation, which occurs between dendrites. The smaller , the more severe the segregation is; When >1, the element exhibits negative segregation, which occurs in the dendrite core.
11. Lines 286 -288 “From Figure 6 and Figure 7, it can be seen that the relation of SDAS and the fraction of dendrite arm area with sampling spots show nearly the same trend but the value of which is varied from S1ij,1 to S1ij,7 and S2ij,1 to S2ij,7.” – not clear.
A: It was changed to“From Figure 6 and Figure 7, it can be seen that the relation of SDAS and the fraction of dendrite arm area with sampling spots show nearly the same trend for different columns, but the relatively value of which is varied from S1ij,1 to S1ij,7 and S2ij,1 to S2ij,7.”
12. Table 3 should be improved ( descriptions of the columns).
A: It was modified.
13. Equation 2 and line 339 “σs is the strength” – what strength, please precise.
A:It is yield strength and was modified now.
14. Equation 3 – what is σ?
A: It was modified now.
15. Equation 4 and Line 351 “Therefore, this paper uses the SDAS (λ2 ) to replace the grain size and bring it into equation (4) to analyze the variation of size of dendrite on hardness.” – Not clear. Please provide the mentioned equation or specify more precisely.
A: Equation (5) was added to clarify this.
16. Line 357 “It can be seen that the experimental data is not in good agreement with the Hall-Petch relationship.” What is presented in Figure 9 – experimental or analytical results? Where is verification of the Hall-Petch relationship What is the basis for this statement?
A: Experimental results were presented in Figure 9. The reason why we draw this conclusion is that,when we pay attention to equation (4) or the newly added equation(5), it should be a liner relationship if the data is in agreement with the Hall-Petch relationship, but what shown in Figure 9 is scattered distribution.
Reviewer 3 Report
Comments and Suggestions for Authors
Researchers studied the homogeneity of CoCrPtB alloy billets using casting and vacuum hot pressing. They explored the relationship between secondary dendrite arm spacing (SDAS) and Vickers hardness/magnetic properties in hot-pressed samples. In the cross-sectional layer, SDAS showed a parabolic trend with a central increase, while the dendrite phase declined. Longitudinally, SDAS and dendrite phase remained uniform, except for an anomaly in the central section. The microstructural inhomogeneity led to variations in material properties. The as-cast alloy exhibited a dendritic microstructure, and hot pressing introduced inhomogeneity. Vickers hardness in dendrite core was consistently lower, decreasing after hot pressing. Magnetic properties, especially coercivity, decreased after hot pressing. Nonhomogeneity in microstructure, hardness, and magnetic properties were attributed to variations in solidification rate and Co phase ratios.
The manuscript is well written and shows interesting results and discussions. Furthermore, it shows potential originality. However, the novelty of the research is not clearly emphasized. Below is a list of suggestive revisions that might help improve the manuscript.
1. What is the main question addressed by the research?
The primary question addressed by the research appears to be an investigation into the microstructural characteristics, mechanical properties (specifically hardness), and magnetic properties of the CoCr13Pt16B10 alloy after undergoing a vacuum hot pressing process. The study aims to understand how the hot pressing treatment influences the alloy's microstructure, hardness, and magnetic performance, providing insights into its potential applications and processing performance.
2. Do you consider the topic original or relevant in the field? Does it address a specific gap in the field?
The topic seems to be both original and relevant in the field. The research delves into the microstructural, mechanical, and magnetic aspects of the CoCr13Pt16B10 alloy following a vacuum hot pressing process. While the specific alloy composition might be unique, the broader investigation into the effects of hot pressing on alloy properties aligns with ongoing research in materials science and metallurgy.
The study appears to address a specific gap in the field by focusing on the impact of vacuum hot pressing on the mentioned alloy. Understanding how this specific processing technique influences the microstructure, hardness, and magnetic properties of the alloy contributes valuable information, especially if there is limited existing research on this particular alloy or the effects of hot pressing on similar compositions.
3. What does it add to the subject area compared with other published material?
The research appears to contribute to the subject area by providing detailed insights into the microstructural, mechanical, and magnetic characteristics of the CoCr13Pt16B10 alloy following a vacuum hot pressing process. The study includes comprehensive analyses, such as XRD patterns, optical and scanning electron micrographs, hardness measurements, and magnetic property evaluations. This wealth of experimental data and thorough examination sets it apart from other published materials.
Comparisons with other published materials would be necessary to make a definitive judgment, but based on the information provided, the study seems to offer a unique and in-depth perspective on the alloy's behavior after hot pressing. The detailed examination of the alloy's microstructure and properties, along with the consideration of various parameters, enhances the understanding of the alloy's response to processing conditions.
4. What specific improvements should the authors consider regarding the methodology? What further controls should be considered?
Replication: Conducting experiments with multiple replicates would strengthen the reliability of the findings, ensuring that observed effects are consistent and not merely due to variability.
Controlled Cooling Rates: Since the solidification rate is identified as a significant factor, controlling and documenting cooling rates during the casting process could provide a more controlled environment for understanding microstructural variations.
In-depth Characterization Techniques: The study mainly employs XRD, OM, SEM, EDS, and hardness measurements. Integrating additional advanced characterization techniques, such as transmission electron microscopy (TEM) or atom probe tomography, could offer even finer details about the microstructure.
Thorough Statistical Analysis: While statistical analysis is mentioned, providing more details on the statistical methods used and potentially employing more advanced statistical tools could enhance the robustness of the conclusions.
Consideration of Alloying Elements: The study focuses on Pt, Cr, and B but does not delve into potential interactions between these elements. Investigating the combined effects of alloying elements on microstructure and properties would provide a more holistic understanding.
Including these aspects would strengthen the methodology and contribute to the overall rigor of the research.
5. Are the conclusions consistent with the evidence and arguments presented and do they address the main question posed?
Yes, the conclusions appear consistent with the evidence and arguments presented throughout the research. The authors effectively link the observed microstructural variations, hardness changes, and magnetic property alterations to the specific processing steps, providing a logical connection between the experimental results and the main research question. The delineation of nonhomogeneities in microstructure, hardness, and magnetic properties after vacuum hot pressing supports the conclusions drawn.
6. Are the references appropriate?
Yes, based on the provided text, it seems like the references are appropriate. The authors refer to various studies and findings to support their statements and provide context to their research. The references cover a range of relevant topics in materials science and metallurgy.
7. Please include any additional comments on the tables and figures.
The tables and figures in the study play a crucial role in presenting the research findings. However, it might be beneficial to enhance the clarity of certain figures by providing more detailed captions or labels. Additionally, the authors could consider using color-coding or annotations to make key points more visually apparent in the figures. Furthermore, a more detailed explanation or interpretation of the tables and figures in the discussion section could help readers better understand the significance of the presented data.
Author Response
Answer for comments 1- 6:Thanks for the suggestions.In the future, more works are need to be done, such as more precise control and the monitoring of cooling rates, thorough statistical analysis with in-depth characterization techniques or in different alloy composition etc.
Answer for comment 7: For the clarity of figures, we had uploaded the high resolution figures with detailed captions or labels, it might be the transformation during editing process which would compress the size of files and the clarity is not good enough. We had some sophisticated color-coding or annotations to make key points visually apparent in the figures. For example, purple and yellow were used in to distinguish the cross-section and longitudinal area specimens, which were shown in most of the figures.
Round 2
Reviewer 2 Report
Comments and Suggestions for Authors
Dear Authors,
My remarks:
1. please check spaces, mainly before references.
2. Line 109 "KV" - in my opinion it should be "kV" (kiloVolts),
3. Line 124 - check commas,
4. Lines 242 - 246 - check grammar "alloy in Figure 1 on the margin.while" and "alloy in Figure 1 in the middle.". In my opinion, the contents are not correct at the moment.
5. The figures are still blurred.
Kind Regards,
Comments on the Quality of English Language
Lines 242 - 246 - check grammar "alloy in Figure 1 on the margin.while" and "alloy in Figure 1 in the middle.". In my opinion, the contents are not correct at the moment.
Author Response
1. please check spaces, mainly before references.
A: it was modified.
2. Line 109 "KV" - in my opinion it should be "kV" (kiloVolts),
A:yes, it should be "kV', already modified.
3. Line 124 - check commas,
A: it was modified.
4. Lines 242 - 246 - check grammar "alloy in Figure 1 on the margin.while" and "alloy in Figure 1 in the middle.". In my opinion, the contents are not correct at the moment.
A: the contens changed as "Figure 4 shows the optical micrographs, the SDAS and fraction of dendrite phase variation of different sampling locations on the margin of the cross-section after hot pressing for CoCr13Pt16B10 alloy in Figure 1 . while Figure 5 shows the same characteristics in the middle of the cross-section for CoCr13Pt16B10 alloy in Figure 1 ."
5. The figures are still blurred.
A: the figures shown in the manuscript is the compressed state, it will be improved during the final editing process if the original high resolution figures are used.